# Discovery of a metabolic alternative to the classical mevalonate pathway

**Nikki Dellas[1,2,3], Suzanne T Thomas[2], Gerard Manning[4]\*[†], Joseph P Noel[1,2]\***

[1]Howard Hughes Medical Institute, Salk Institute for Biological Studies, La Jolla, United States; [2]Jack H Skirball Center for Chemical Biology and Proteomics, Salk Institute for Biological Studies, La Jolla, United States; [3]Department of Chemistry and Biochemistry, University of California, San Diego, La Jolla, United States; [4]Razavi Newman Center for Bioinformatics, Salk Institute for Biological Studies, La Jolla, United States

**Abstract** Eukarya, Archaea, and some Bacteria encode all or part of the essential mevalonate (MVA) metabolic pathway clinically modulated using statins. Curiously, two components of the MVA pathway are often absent from archaeal genomes. The search for these missing elements led to the discovery of isopentenyl phosphate kinase (IPK), one of two activities necessary to furnish the universal five-carbon isoprenoid building block, isopentenyl diphosphate (IPP). Unexpectedly, we now report functional IPKs also exist in Bacteria and Eukarya. Furthermore, amongst a subset of species within the bacterial phylum Chloroflexi, we identified a new enzyme catalyzing the missing decarboxylative step of the putative alternative MVA pathway. These results demonstrate, for the first time, a functioning alternative MVA pathway. Key to this pathway is the catalytic actions of a newly uncovered enzyme, mevalonate phosphate decarboxylase (MPD) and IPK. Together, these two discoveries suggest that unforeseen variation in isoprenoid metabolism may be widespread in nature.

**\*For correspondence:**
manning@manninglab.org (GM);
noel@salk.edu (JPN)

[†]**Present address:** Department of Bioinformatics and Computational Biology, Genentech, San Francisco, United States

**Competing interests:** The authors declare that no competing interests exist.

**Reviewing editor**: Detlef Weigel, Max Planck Institute for Developmental Biology, Germany

## Introduction

Isoprenoids constitute a substantial family of primary and secondary metabolites in all three domains of life. These molecules play essential and specialized roles for their hosts including modulation of membrane fluidity (cholesterol, hopanoids, squalene), chemical defense and communication (mono-, sesqui- and diterpenes), photoprotection and energy transfer (carotenoids) (*Lu and Li, 2008*), and growth regulation (giberellins) (*Hedden and Kamiya, 1997*). Additionally, isoprenoids such as quinones, chlorophyll, bacteriochlorophyll, and some cellular proteins are tethered to a polyisoprenoid chain to direct localization to membranes or to potentiate interactions with other biomolecules (*Nowicka and Kruk, 2010*; *Schafer and Rine, 1992*).

Isopentenyl diphosphate (IPP, **1**) and its isomer, dimethylallyl diphosphate (DMAPP, **2**), are the five-carbon building blocks of all higher order isoprenoids. It is currently accepted that IPP is biosynthesized via one of two metabolic pathways: the mevalonate (MVA or sometimes MEV) pathway (*Katsuki and Bloch, 1967*; *Lynen, 1967*) or the 1-deoxy-D-xylulose 5-phosphate (DXP) pathway (also known as the 2-C-methyl-D-erythritol 4-phosphate, MEP, or Rohmer pathway) (*Arigoni et al., 1997*; *Eisenreich et al., 1998*; *Rohmer, 1999*). These two metabolic systems utilize non-homologous enzymes that evolved independently to produce the same 5-carbon end product, IPP (**1**). Typically, a given organism uses one pathway or the other (*Lange et al., 2000*). Eukarya appear to encode the classical MVA pathway (with some exceptions, see [*Cassera et al., 2004*]); plants additionally encode the DXP pathway that operates in the chloroplast. Most bacteria employ the DXP pathway with the exception of several phyla that contain full or partial MVA pathway genes (*Bochar et al., 1999*, *Lombard and Moreira, 2011*).

**eLife digest** Living things make thousands of chemicals that are vital for life, and are also useful as medicines, perfumes, and food additives. The largest family of these natural chemicals is called the isoprenoids, and members of this family are found in all three domains of life: the eukaryotes (such as plants and animals), the Archaea (an ancient group of single-celled microbes), and bacteria.

The isoprenoids are made from a smaller building block called isopentenyl diphosphate, IPP for short, that contains five carbon atoms and two phosphate groups. IPP can be produced in two ways. The classical mevalonate pathway is found in most eukaryotes, including humans; statin drugs are used to inhibit this pathway to treat those with high cholesterol and reduce the risk of heart disease. The second pathway does not use the compound mevalonate and is found in many, but not all, bacteria as well as the chloroplasts of plants. Until recently, however, the enzymes needed for the last two steps of the classical mevalonate pathway appeared to be missing in the Archaea and in some bacteria.

Researchers subsequently discovered that an enzyme called isopentenyl phosphate kinase, shortened to IPK, was responsible for one of these two missing steps—the addition of IPP's second phosphate group. The way this enzyme worked also suggested that there was an alternative mevalonate pathway in which the order of the last two steps was reversed. However, the identity of the enzyme responsible for the other step—the removal of a molecule of carbon dioxide to make the starting material needed by IPK—remained mysterious.

Now Dellas et al. have discovered the enzyme responsible for this missing step in Green non-sulphur bacteria, confirming the existence of the alternative mevalonate pathway for the first time. Previously it had been thought that this enzyme acted in the classical mevalonate pathway; but in fact this enzyme has evolved a new function and is not involved in the classical pathway at all. Moreover, Dellas et al. show that Green non-sulphur bacteria, and some eukaryotes, also have functional IPK enzymes. This means that IPK has now unexpectedly been observed in all three domains of life, and hints at another target to medically control mevalonate pathways. The discovery of the missing enzyme in the alternative pathway opens the door to the re-examination of many other living things, to find which have the new pathway and to work out why.

While the domain Archaea and the bacterial class Chloroflexi do not encode gene homologs for the DXP pathway, nearly all of these species characterized genomically to date encode an incomplete classical MVA pathway. That is, many of these species lack identifiable genes encoding enzymes for one or both of the terminal steps of IPP biosynthesis through mevalonate. This observation is puzzling due to the essentiality of isoprenoid compounds in all organisms. Recent phylogenetic and experimental data suggest that Archaea (and probably also the Chloroflexi) encode an alternative MVA pathway, which bifurcates from the classical pathway following the phosphorylation of mevalonate to phosphomevalonate (MVAP, also known as mevalonate 5-phosphate, **3**) (*Grochowski et al., 2006*; *Matsumi et al., 2010*; *Miziorko, 2010*) (*Figure 1*). Following MVAP biosynthesis, the classical MVA pathway uses phosphomevalonate kinase (PMK) to phosphorylate MVAP (**3**) producing diphosphomevalonate (MVAPP, also known as mevalonate 5-diphosphate, **5**). MVAPP then undergoes a phosphorylation-dependent decarboxylation catalyzed by mevalonate 5-diphosphate decarboxylase (MDD, also known as MDC or DPM-DC) producing the key five-carbon isoprenoid building block, IPP (**1**).

The alternative MVA pathway is posited to reverse these steps; decarboxylation of MVAP (**3**) followed by phosphorylation of isopentenyl phosphate (IP, **4**) (*Figure 1*). The first step would require a currently undetected enzyme activity, MVAP decarboxylase (MPD). The final phosphorylation activity has been characterized in Archaea and is catalyzed by isopentenyl phosphate kinase (IPK), an ATP-dependent kinase that phosphorylates IP (**4**) forming IPP (**1**) (*Figure 1*).

Until now, the *IPK* gene has not been studied among the Chloroflexi bacteria or within eukaryotic organisms presumably encoding all the enzymes of the classical MVA pathway. We performed extensive phylogenetic analyses to reveal a spotty distribution of IPK-bearing organisms in animals, several fungi, all plants, and some bacteria (including those from the class Chloroflexi). Following heterologous expression, purification, and kinetic characterization of several proteins encoding representative *IPK*-like genes from each domain of life, we show that fully active and highly specific IPKs unexpectedly exist outside of Archaea.

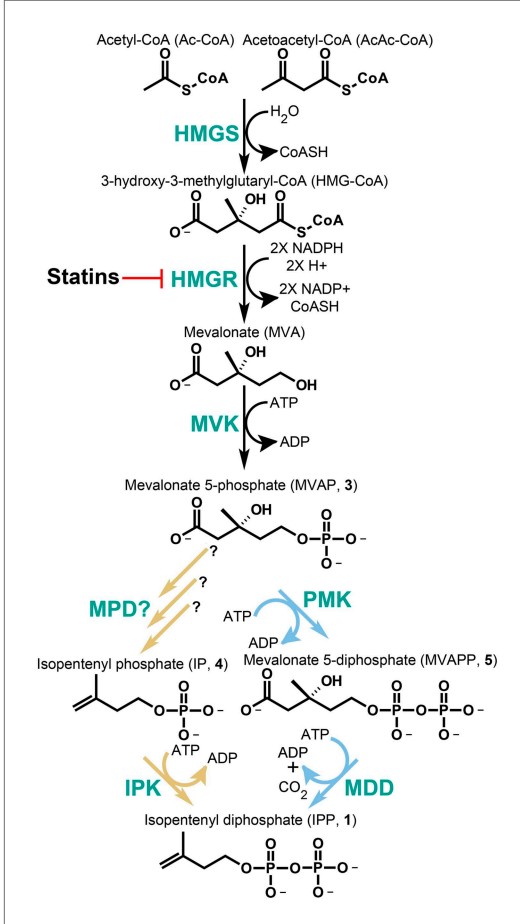

**Figure 1**. The classical and alternative MVA pathways. Both branches of the MVA pathway begin with acetyl-CoA (and acetoacetyl-CoA) and proceed through a series of enzymatic reactions involving 3-hydroxy-3-methylglutary-CoA Synthase (HMGS), 3-hydroxy-3-methylglutary-CoA Reductase (HMGR, the presumed early rate-limiting step), and mevalonate kinase (MVK) before branching. At the bifurcation, the canonical MVA pathway, highlighted by light blue arrows, guides MVAP (**3**) through an additional phosphorylation reaction followed by a phosphorylation-dependent decarboxylation carried out by phosphomevalonate kinase (PMK) and diphosphomevalonate decarboxylase (MDD), respectively. The alternative MVA pathway, highlighted with light brown arrows, hypothetically decarboxylates MVAP (**3**) prior to the phosphorylation reaction carried out by IPK but the former step has not been discovered until now (*Grochowski et al., 2006*). The enzymes MVK, PMK, MDD, and IPK all consume ATP during catalysis. All enzymes are shown in green type. Statins serve as inhibitors of HMGR as highlighted.

One such fully active IPK was characterized from the Chloroflexi bacteria, *Roseiflexus castenholzii*. Interestingly, by classical genome annotation, the Chloroflexi bacteria, as well as certain other archaeal and bacterial groups, contain only a subset of components of the classical (PMK-, MDD-utilizing) and alternative (MPD-, IPK-utilizing) MVA pathways, resulting in what appears to be the existence of two incomplete branches of the MVA pathway in one organism (*Figure 1*). For example, Chloroflexi bacteria appear to be missing the *PMK* gene from the classical pathway and the *MPD* gene from the alternative pathway (*Lombard and Moreira, 2011*). This arrangement contrasts with archaeal species, which do not appear to encode either PMK or MDD from the classical MVA pathway.

Through biochemical characterization of the enzymes encoded by the remaining MVA pathway genes (*MDD* and *IPK*) from the Chloroflexi bacterium, *R. castenholzii*, we unequivocally demonstrate that a functional alternative MVA pathway does exist and is encoded in a completely unexpected manner. In *R. castenholzii*, the putative MDD of the classical MVA pathway surprisingly is fully functional, and until now, expresses an unknown MPD activity associated with the alternative MVA pathway. Combined with our observation of a fully functional *IPK*-like gene from this same organism, these results provide the first definitive bioinformatic and experimental evidence for a fully operative alternative MVA pathway in nature.

## Results

### Phylogenetic analyses of IPK homologs

IPK is a member of the amino acid kinase (AAK) superfamily. To date, the search for eukaryotic *IPK*-like genes has been limited since putative eukaryotic IPK homologs are divergent in sequence and can be difficult to distinguish from other AAK superfamily genes (*Lombard and Moreira, 2011*). Within IPK, we previously identified a catalytically essential active site histidine residue that is not conserved among AAK family members belonging to the amino acid kinase superfamily (*Dellas and Noel, 2010*). Using this structurally and functionally defined residue as an indication of possible IPK activity (*Dellas and Noel, 2010*), we next identified putative IPK homologs from all three domains of life. We used PSI-BLAST and profile Hidden Markov Models (HMMs) to detect IPK homologs in public protein, expressed sequence tag (EST), and genome databases. IPK homologs bearing the key histidine signature appear in nearly all archaea (*Lombard and Moreira, 2011*, *Matsumi et al., 2010*), a cluster of Chloroflexi bacteria (*Lombard and*

*Moreira, 2011*), every sequenced green plant genome, and, in an exceptionally sporadic distribution, across most major eukaryotic lineages (*Figure 2*, *Table 1*).

IPK appears to have been lost independently in many animal lineages (*Figure 2*, *Table 1*). It is absent from choanoflagellates and sponges, but found in early branching animals such as *Trichoplax*, cnidarians (*Nematostella*, but not *Hydra*), and corals. It is found in bilaterians, including molluscs (*Aplysia sp.*, *Lottia gigantea*), annelids (earthworm and leech), and a crustacean (lobster), but not in any

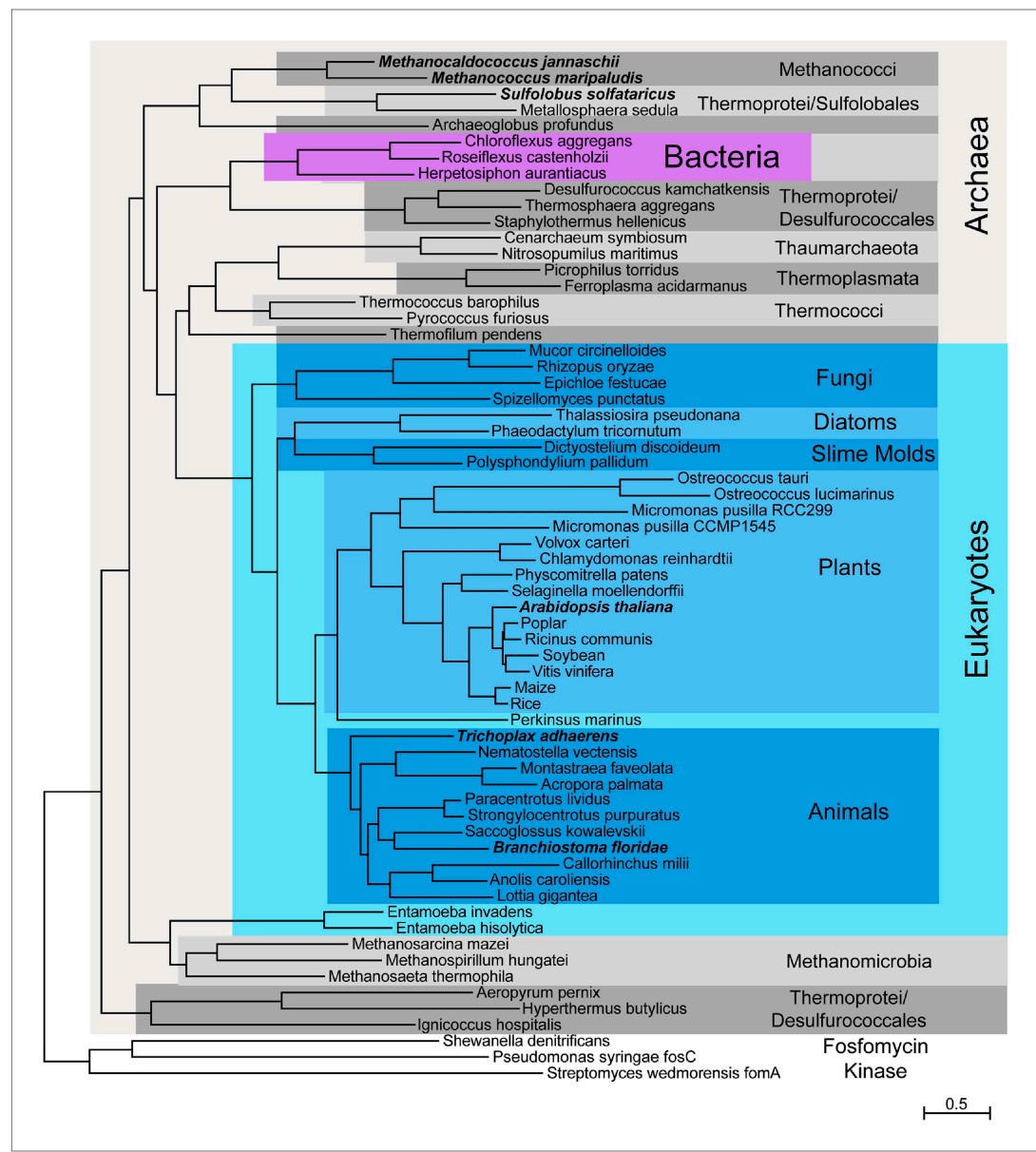

**Figure 2**. Phylogenetic distribution of IPK across the three domains of life. Maximum likelihood tree of selected IPK protein sequences. Eukaryotes are highlighted with blues, selected archaeal clades with grays and a small group of bacteria with purple. The tree is anchored by several bacterial fosfomycin kinases. See *Figure 2—source data 1* for an alignment of IPK homologs. See *Figure 2—source data 2* for a table of IPK homolog sequences.
The following source data are available for figure 2:

**Source data 1**. Alignment of IPKs from the three domains of life.
**Source data 2**. Additional IPK sequences from an assortment of sequence databases.

Biochemistry | Genomics and evolutionary biology

**Table 1.** Phylogenetic distribution of IPK in Eukarya

| Classification | IPK-bearing species | Species with no IPK |
|---|---|---|
| Mammals | | 35 mammalian genomes |
| Birds | | *Gallus gallus* |
| | | *Taeniopygia guttata* |
| | | *Meleagris gallopavo* |
| Reptiles | *Anolis carolensis* | |
| | *Philodryas olfersii* (EST)* | |
| Amphibians | *Notophthalmus viridescens* (EST) | *Xenopus tropicalis* |
| Fish | *Callorhinchus milii* (partial IPK from draft genome) | *Danio rerio* |
| | | *Oryzias latipes* |
| | | *Takifugu rubripes* |
| | | *Tetraodon nigrovidris* |
| | | *Gasterosteus aculeatus* |
| Invertebrate chordates | *Branchiostoma floridae* | *Ciona intestinalis* |
| | *Saccoglossus kowalevskii* | *Ciona savignyi* |
| Echinoderms (Deuterostomes) | *Paracentrotus lividus* (EST) *(almost complete assembly)* | |
| | *Strongylocentrotus purpuratus (almost complete prediction)* | |
| Arthropods | *Homarus americanus* (EST) | *Drosophila sp.* (**Cassera et al., 2004**) |
| | | *Apis mellifera* |
| | | *Aedes aegypti* |
| | | *Anopheles gambiae* |
| | | *Culex quinquefasciatus* |
| | | *Pediculus humanus* |
| | | *Ixodes scapularis* |
| | | *Daphnia pulex* |
| Other bilaterians | *Hirudo medicinalis* (EST) | *Capitella teleta* |
| | *Eisenia fetida* (EST) | *Helobdella robusta* |
| | *Lottia gigantea* | *Schistosoma mansoni* |
| | *Aplysia kurodai* (EST) | *Schistosoma japonicum* |
| | *Aplysia californica (partial prediction)* | *Brugia malayi* |
| | | *Meloidogyne incognita* |
| | | *Pristionchus pacificus* |
| Cnidarians | *Acropora palmate* (EST) | *Hydra magnapilliata* |
| | *Montastraea faveolata* (EST) | |
| | *Nematostella vectensis (predicted)* | |
| Other early metazoans | *Trichoplax adherens* | *Amphimedon queenslandica* |
| Pre-metazoans (within Holozoa) | | *Monosiga brevicollis* |
| | | *Salpingoeca rosetta* |
| | | *Capsaspora owczarzaki* |
| Fungi | *Spizellomyces punctatus* | 34 fungal genomes |
| | *Rhizopus oryzae* | |
| | *Mucor circinelloides* | |
| | *Epichloe festucae* (EST)‡ | |

*Table 1. Continued on next page*

Table 1. Continued

| Classification | IPK-bearing species | Species with no IPK |
|---|---|---|
| Green plants | all plants, including: | |
| | *Selaginella moellendorffii* | |
| | *Physcomitrella patens* | |
| | *Ostreococcus tauri* | |
| | *Ostreococcus lucimarinus* | |
| | *Micromonas pusilla* | |
| | *Chlamydomonas reinhardtii* | |
| | *Volvox carteri* | |
| Amoebozoa | *Dictyostelium discoideum* | |
| | *Polysphondylium pallidum* | |
| | *Entamoeba hislolytica*§ | |
| | *Entamoeba invadens*§ | |
| | *Entamoeba dispar*§ | |
| Alveolata | *Perkinsus marinus* | *Plasmodium* sp. (**Lynen, 1967**) |
| | | *Cryptosporidium* sp. (**Hedden and Kamiya, 1997**) |
| | | *Tetrahymena thermophila* |
| | | *Paramecium tetraurelia* |
| | | *Ichthyophthirius multifilis* |
| Diatoms | *Thalassiosira pseudonana* | |
| | *Phaeodactylum tricornutum* | |
| Kinetoplastida | | *Trypanosoma* sp. (**Chew and Bryant, 2007**) |
| | | *Leishmania* sp. (**Chew and Bryant, 2007**) |
| Excavates | | *Giardia lamblia* |
| | | *Trichomonas vaginalis* |
| Others | *Malawimonas jakobiformis* (EST) | *Naegleria gruberi* |
| | *Alexandrium tamarense* | |

*EST stands for expressed sequence tag. All other IPKs were found in draft or complete genomes.
‡Since IPK was not found in related fungal genomes, this may be a case of horizontal transfer or even EST library contamination.
§Greater similarity to IPK from a clade of archaea than to eukaryotes.

insect or nematode. Within deuterostomes, it is found in the sea urchin and sea star, as well as the hemichordate *Saccoglossus kowalevskii* (acorn worm) and the chordate *Branchiostoma floridae* (lancelet). Within the vertebrates, it is found in a shark (*Callorhinchus milii*) but no teleost fish; in an amphibian (the newt *Notophthalmus viridescens*) but not in frogs; and in a lizard (*Anolis carolinensis*) and a snake (*Philodryas olfersii*), but not in any bird or mammal. Within the fungi, IPK is present in one of two sequenced chytrids and two other basal fungal genomes, but is otherwise absent.

While all identified IPK homologs retain 11 invariant residues (see *Table 2*), their sequences are otherwise highly divergent, and their predicted phylogeny agrees only in part with organismal taxonomy. For instance, while all plant and animal IPKs cluster together, sequences from the archaeal class Thermoprotei fall into four distinct clusters on the calculated phylogenetic tree, and in the eukaryotic genus Entamoeba, sequences cluster with the archaeal class Methanomicrobia (*Figure 2*). This sporadic grouping pattern suggests horizontal gene transfer and/or parallel evolution is not uncommon and may relate to shared organismal needs in similar ecological niches.

**Table 2.** Conserved residues in IPK

| Highly-conserved residues* | Partially conserved residues |
|---|---|
| K6, G8, G9, K15, G54, H60, P140, G144, D213, T215, G216 | K221†, G253‡, T254‡ |

*Numbering is in accordance with IPK from *M. jannaschii*.
†Conserved in nearly all species except one possibly due to a gene prediction error.
‡May be invariant, but mis-predicted in several sequences.

## Kinetic characterization of IPKs, PMKs, and MDDs

To confirm the phylogenetic analyses and identification of putative *IPK* genes, we next overexpressed, purified, and biochemically characterized seven IPK homologs selected from all three domains of life. These include the previously characterized IPK from *M. jannaschii* (**Grochowski et al., 2006**; **Dellas and Noel, 2010**) and homologs from two other archaeal species, *Methanococcus maripaludis* and *Sulfolobus solfataricus* (one of the few archaeal species that also encodes the complete classical MVA pathway), the bacterium *Roseiflexus castenholzii* (an organism with an annotated MDD but no obvious PMK), and three eukaryotes, *Trichoplax adhaerens* (early-branching metazoan), *Branchiostoma floridae* (chordate), and *Arabidopsis thaliana* (plant). Kinetic experiments were performed using a quantitative lactate dehydrogenase-pyruvate kinase coupled assay demonstrating that all seven IPK homologs catalyze the efficient phosphorylation of IP (**4**) to IPP (**1**) with high specificity as reported in *Table 3*.

These IPK-bearing organisms all appear from previous annotations to encode genes associated with the classical MVA pathway. To explore possible MVA pathway bifurcation, we next overexpressed, purified, and biochemically characterized PMK-like and MDD-like proteins from organisms containing biochemically verified IPKs. PMKs from *B. floridae* and *A. thaliana* catalyze the efficient phosphorylation of MVAP as annotated (**3**) with similar kinetic constants (*Table 3*). PMK from *S. solfataricus* is catalytically active using ATP and MVAP (**3**) as substrates, however, its MDD catalyzes the phosphorylation-dependent decarboxylation of MVAPP (**5**) to IPP (**1**) at a very slow rate. Under steady state conditions, we were unable to accurately determine kinetic parameters for *S. solfataricus* MDD.

## Kinetic characterization of a Chloroflexi MPD

Unexpectedly, the MDD homolog from *R. castenholzii* does not catalyze the decarboxylation of MVAPP (**5**) to IPP (**1**) at a measurable rate as annotated. Instead, it efficiently catalyzes the phosphorylation-dependent decarboxylation of MVAP (**3**) to IP (**4**) (*Table 3*). This annotated *R. castenholzii* MDD, in fact, functions in a completely unexpected way as a bona fide MPD assuming the metabolic role of the long sought MPD of the alternative MVA pathway.

The determined catalytic constants of *R. castenholzii* MPD with MVAP suggest it is less catalytically efficient than bona fide MDD enzymes (*Table 3*). *R. castenholzii*, however, is a thermophilic bacterium and assays at its optimal growth temperature (50°C and above) (**Hanada et al., 2002**) were not possible due to stability limitations of the coupled assay system. Fluorescence thermal shift assays (**Pantoliano et al., 2001**) demonstrate that recombinant *R. castenholzii* MPD is stable at temperatures exceeding 90°C (inflection point of the slope of the fluorescence vs temperature curves ($T_m$) = 94°C at pH 5.8), indicating that the enzyme maintains structural integrity at temperatures well beyond what it commonly encounters in nature and is most likely maximally active at high temperatures (*Figure 3*).

To the best of our knowledge, this is the first enzyme discovered to function as a catalytically efficient MPD. Given the annotation bias associated with functional assignment based upon sequence homology alone, this newly discovered MPD activity encoded by an MDD-like sequence suggests that functional expectations of the enzymes comprising the canonical MVA biosynthetic pathway must be reexamined.

## GC-MS quantification of reconstituted alternative and classical MVA pathways

To confirm that the functional *R. castenholzii* MPD (MDD-like protein sequence) catalyzes the transformation of MVAP (**3**) to IP (**4**) in a pathway specific manner with a measurable flux, we next reconstituted each pathway in vitro using a five-enzyme reaction mixture of homogenously purified enzymes and substrates. Briefly, MVAP (**3**) and ATP were incubated for 30 min with MPD and IPK from *R. castenholzii*. The reaction mixture was then diluted into a mixture containing all necessary downstream enzymes to

**Table 3.** Steady-state kinetic constants for enzymes of the MVA pathway*

| Organism | gi number | Enzyme | Substrate | $K_M$ (µM) | $k_{cat}$ (s$^{-1}$) | $K_i$ (µM) | $k_{cat}/K_M$ (s$^{-1}$µM$^{-1}$) | R$^2$ (†) |
|---|---|---|---|---|---|---|---|---|
| *M. jannaschii* | 15668214 | IPK | IP | 4.3 (± 0.6)‡ | 1.46 (± 0.03) | –§ | 0.34 (± 0.05) | 0.90 |
| *M. maripaludis* | 132664414 | IPK | IP | 21.4 (± 4.3) | 15.2 (± 1.4) | 877 (± 550) | 0.71 (± 0.16) | 0.99 |
| *S. solfataricus* | 15897030 | IPK | IP | 23.6 (± 4.8) | 0.91 (± 0.05) | – | 0.04 (± 0.01) | 0.92 |
| | 15899698 | PMK | MVAP | 23.1 (± 3.8) | 1.98 (± 0.10) | 5500 (± 2200) | 0.09 (± 0.01) | 0.98 |
| *R. castenholzii* | 156743980 | IPK | IP | 4.3 (± 0.7) | 1.70 (± 0.04) | – | 0.40 (± 0.06) | 0.96 |
| | 156740939 | MPD# | MVAP | 152 (± 38) | 1.7 (± 0.1) | – | 0.011 (± 0.003) | 0.98 |
| | 156740939 | MDD | MVAPP | ND# | ND | ND | ND | ND |
| *B. floridae* | NA¶ | IPK | IP | 13.3 (± 2.0) | 27.2 (± 1.2) | 2820 (± 1700) | 2.05 (± 0.32) | 0.98 |
| | 260829481 | PMK | MVAP | 38.0 (± 7.3) | 12.6 (± 0.4) | – | 0.33 (± 0.06) | 0.93 |
| | 260794527 | MDD | MVAPP | 15.1 (± 3.7) | 1.36 (± 0.09) | – | 0.09 (± 0.02) | 0.89 |
| *A. thaliana* | 22329798 | IPK | IP | 0.79 (± 0.35) | 1.9 (± 0.2) | 522 (± 381) | 2.4 (± 1.1) | 0.95 |
| | 15222502 | PMK | MVAP | 11.8 (± 2.0) | 20.9 (± 0.7) | – | 1.77 (± 0.31) | 0.97 |
| | 15224931 | MDD | MVAPP | 15.7 (± 5.0) | 2.02 (± 0.13) | – | 0.13 (± 0.04) | 0.89 |
| *T. adhaerens* | 195996013 | IPK | IP | 3.1 (± 1.6) | 2.4 (± 0.2) | – | 0.77 (0.40) | 0.79 |

*See **Table 3—source data 1**, **Table 3—source data 2**, **Table 3—source data 3**, and **Table 3—source data 4** for Michaelis-Menten fitted kinetic curves for each enzyme.

†R2 represents goodness of fit for the kinetic curve to the experimental data. R2 = 1.0 − (SS$_{reg}$/SS$_{tot}$), where SS$_{reg}$ = sum of squares, SS$_{tot}$ = sum of squares of the distances between each point and a horizontal line passing through the average of all y values.

‡Values in parentheses represent standard error (or propagation of error) for each calculated kinetic constant.

§$K_i$ constant was not calculated or was not applicable.

#Not detected.

¶*B. floridae* IPK sequence was predicted from scaffold_167 of the genome assembly (**Putnam et al., 2008**) using genewise and manual annotation.

The following source data are available for table 3:

**Source data 1**. Steady-state kinetic plots for IPKs listed above each curve.

**Source data 2**. Steady-state kinetic plots for PMKs listed above each curve.

**Source data 3**. Steady-state kinetic plots for MDDs listed above each curve.

**Source data 4**. Stead-state kinetic plots for MPD from *R. castenholzii*.

biosynthesize a higher order isoprenoid metabolite, namely the GC-detectable sesquiterpene 5-*epi*-aristolochene (5-EA, **7**). Similar reactions were performed for the in vitro reconstitution of the classical MVA pathway using PMK and MDD from each respective organism.

The in vitro classical MVA pathway reactions for *S. solfataricus*, *A. thaliana*, and *B. floridae* yield 39%, 49%, and 52% conversion of MVAP (**3**) to 5-EA (**7**), respectively (**Figure 4**). The in vitro classical MVA pathway reaction from *R. castenholzii* (using *B. floridae* PMK to complete the missing step) yields less then 0.2% conversion to 5-EA (**Figure 4**). In contrast, the in vitro alternative MVA pathway from *R. castenholzii* enzymes yields a 2.0% conversion of MVAP (**3**) to 5-EA (**7**). *B. floridae*, *A. thaliana*, and *S. solfataricus* were tested for a possible alternative MVA pathway using their IPKs and MDDs. These reactions do not yield any detectable 5-EA (**7**), indicating that these MDDs behave as expected from their gene annotations and lack detectable MPD activity. Control reactions lacking ATP, enzyme, or MVAP (**3**) do not yield any 5-EA (**7**).

## Modeling of *R. castenholzii* MPD reveals an altered MVAP (P) selectivity filter

The experiments described above indicate that the MDD-like enzyme from *R. castenholzii* does not accept MVAPP (**5**) as a substrate. Instead, *R. castenholzii* 'MDD' functions as an MPD and possesses a newly discovered biochemical activity, namely the utilization of MVAP (**3**) as a substrate for the phosphorylation-dependent decarboxylation of MVAP (**3**) to IP (**4**). These biochemical results conclusively

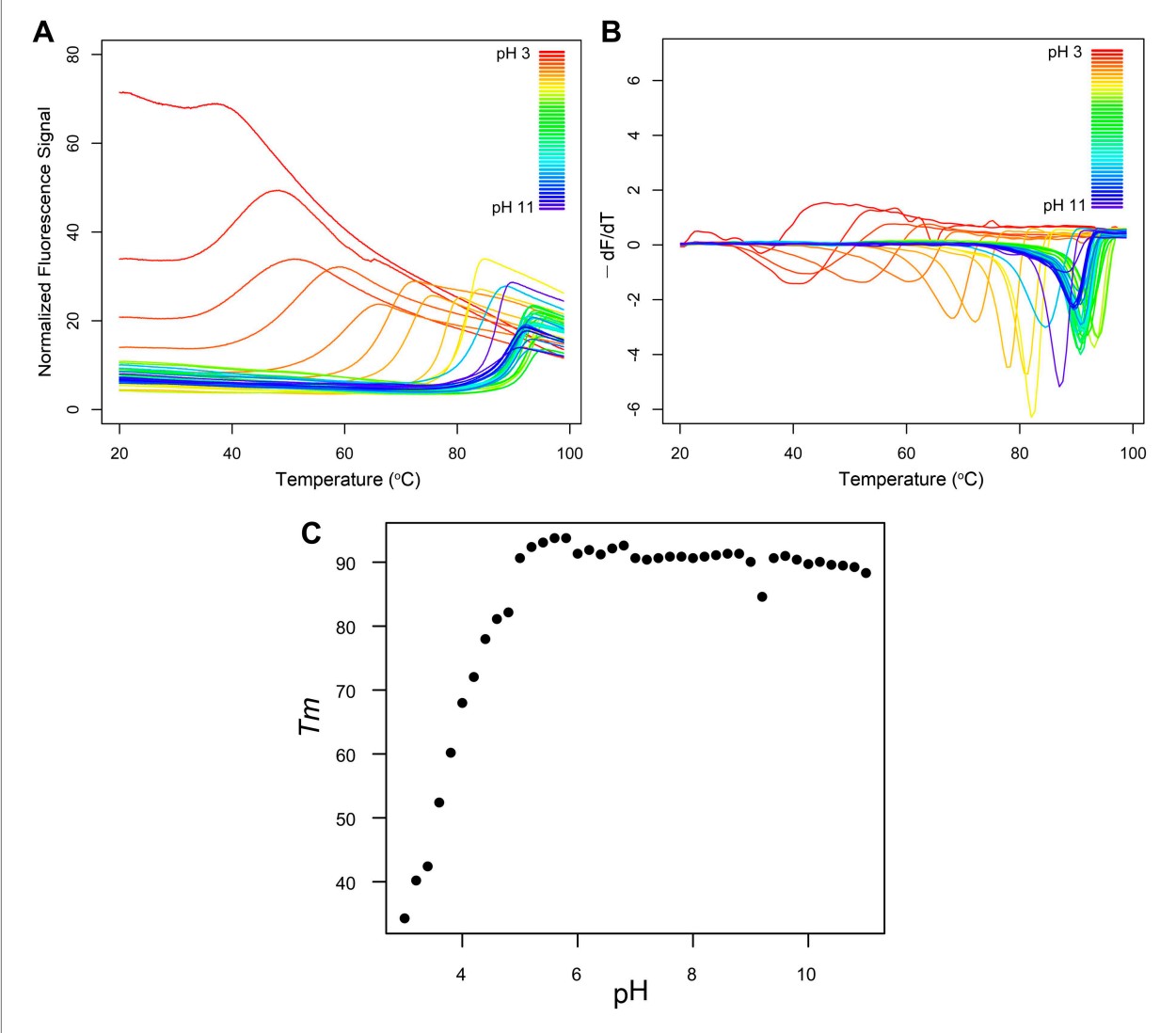

**Figure 3**. Fluorescence thermal shift assays of *Roseiflexus castenholzii* MDD-like MPD. (**A**) Thermograms for *R. castenholzii* MPD in 100 mM buffer (pH 3.0–3.8 citric acid; pH 4.0–4.8 sodium acetate; pH 5.0–5.8 sodium citrate; pH 6.0–6.8 sodium cacodylate; pH 7.0–7.8 sodium HEPES; pH 8.0–8.8 Tris-HCl; pH 9.0–11.0 CAPSO) colored from red to violet (acidic to alkaline pH depicted in the inset). (**B**) Negative derivatives of the thermograms (−dF/dT) color-coded as in (**A**). (**C**) $T_m$s for each of the curves show in (**A**) and (**B**) plotted as a function of pH. *R. castenholzii* MPD was unfolded from pH 2.2 to 2.8 at 20°C (data now shown).

demonstrate that this *MDD*-like gene in fact encodes an MPD. Closer examinations of sequences and three-dimensional structures of MDDs highlight catalytic elements that set MPD from *R. castenholzii* apart from other bacterial MDDs. A structural model of *R. castenholzii* MPD was computed using Swissmodel (*Arnold et al., 2006*; *Kiefer et al., 2009*) and the *Staphylococcus aureus* MDD structure (PDBID 2HK2) as a template (26% sequence identity to *R. castenholzii* MPD) (*Byres et al., 2007*).

Residues surrounding the diphosphate group of 6F-MVAPP (**5**) bound to *S. epidermidis* MDD (PDBID 3QT7(*Barta et al., 2011*)) were then contrasted with the corresponding residues in the *R. castenholzii* MPD model. A comparison of the phosphate-binding residues of the canonical MDD from *S. epidermidis* to the same positions in *R. castenholzii* MPD reveals that the *R. castenholzii* MPD lacks several conserved hydrogen bonding interactions associated with recognition of the terminal phosphate on the diphosphate moiety of MVAPP (**5**) (*Figure 5*). Second, the *R. castenholzii* MPD contains additional residues including arginine and lysine side chains that appear poised to bind the single phosphate of MVAP (**3**). Finally, these residues, unique to atypical MDDs such as the *R. castenholzii* MPD, clash sterically and electronically with the terminal phosphate on the diphosphate moiety of MVAPP (**5**) (*Figure 5*, *Table 4*).

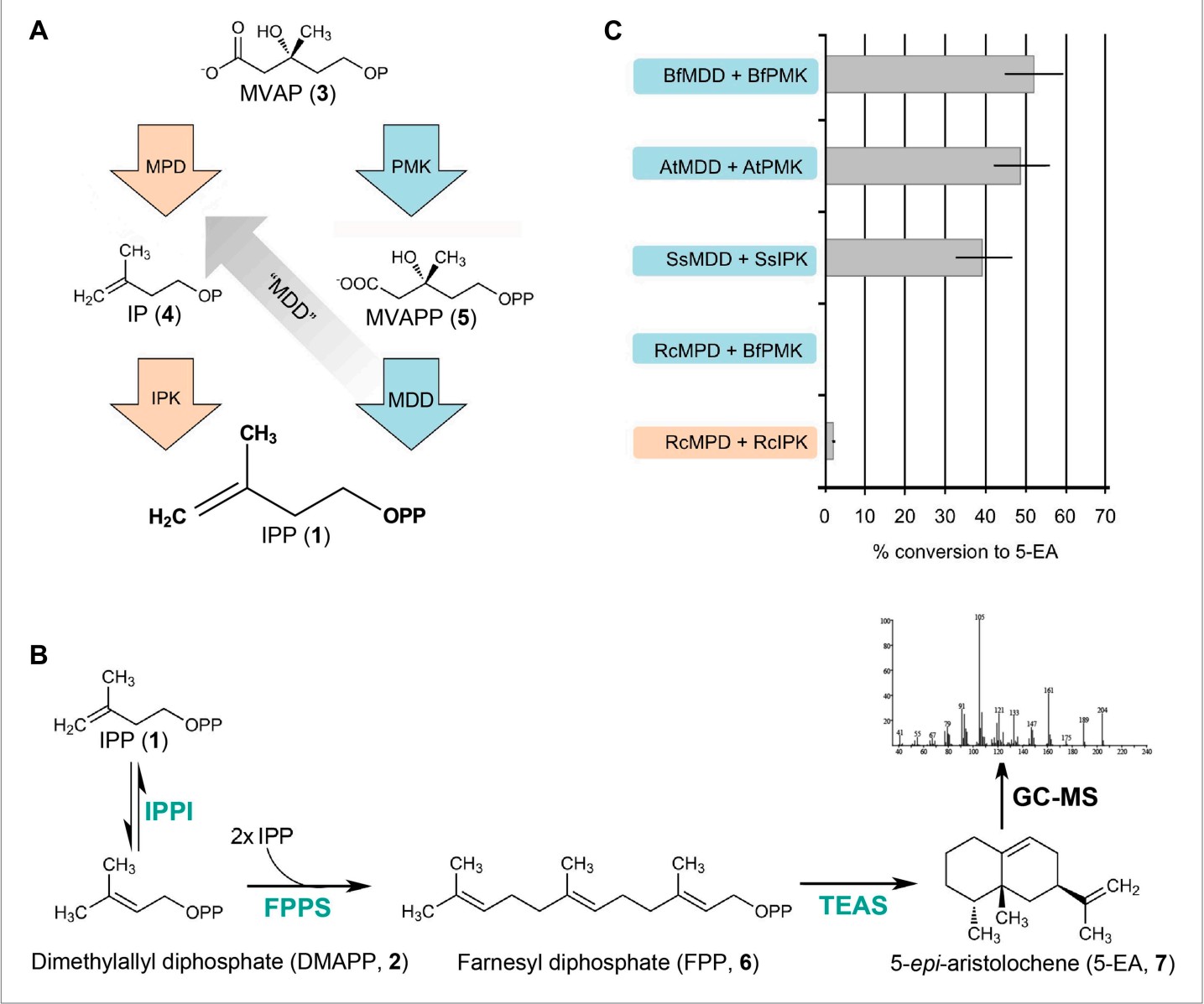

**Figure 4**. In vitro reconstitution and GC-MS analysis of the alternative and classical MVA pathways. (**A**) Terminal steps of the MVA pathways shown with enzymes of the alternative (orange) and classical (blue) pathways depicted as arrows. The putative neofunctionalization of MDD to MPD is highlighted by a grey arrow. (**B**) In vitro assays include either the alternative or the classical enzymes to produce IPP (**1**) (as shown in panel **A**) as well as all downstream enzymes, including isopentenyl phosphate isomerase (IPPI) to produce DMAPP (**2**), farnesyl diphosphate synthase (FPPS) to produce farnesyl diphosphate (FPP, **6**) and tobacco 5-*epi*-aristolochene synthase (TEAS) to produce the sesquiterpene product, 5-EA (**7**). Products are separated by GC and detected by MS ionization and fragmentation. Enzymes used are highlighted in turquoise type. (**C**) Results of the in vitro reconstitutions of various enzyme combinations. The y-axis of the graph represents combinations of enzymes shown in panel **A** and panel **B** and the % conversion to the expected sesquiterpene end product, 5-EA (**7**) shown as grey bars along the x-axis. Abbreviations of organisms are as follows: Bf = *Branchiostoma floridae*, At = *Arabidopsis thaliana*, Ss = *Sulfolobus solfataricus*, and Rc = *Roseiflexus castenholzii*. Note, RcMPD is colored both orange and blue on the y-axis, depending on whether it is being tested as an MDD (blue) or an MPD (orange).

Phylogenetic analyses demonstrate that MDDs from the Chloroflexi are distant from those of other bacteria but similar to those of the two archaeal classes that also encode an IPK but no PMK, the Haloarchaea and Thermoplasmata classes. The model described above thus illustrates a logical route through which atypical MDDs may have over time attenuated MVAPP (**5**) substrate selectivity and acquired high selectivity for MVAP (**3**).

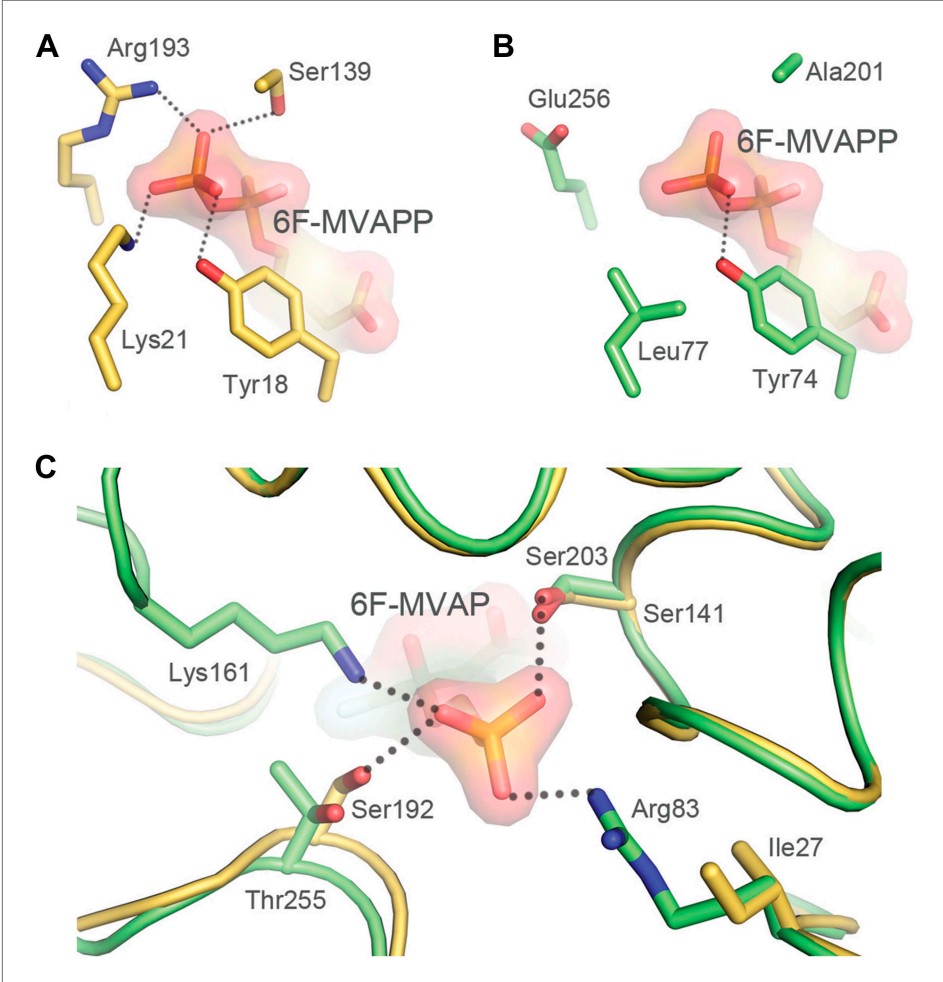

**Figure 5**. Structural comparisons of a bona fide MDD and MPD. Interactions between the terminal phosphate of 6F-MVAPP and the surrounding amino acid residues from the crystal structure of *S. epidermidis* MDD (PDB ID 3QT7(**Barta et al., 2011**)) and the 3D model of MPD from *R. castenholzii*. (**A**) *S. epidermidis* MDD has multiple interactions with the diphosphate of MVAPP (**5**). Atoms are colored by type with carbon gold. (**B**) The active site model of *R. castenholzii* MPD lacks many of the key interactions shown by *S. epidermidis* MDD in panel **A**. Atoms are colored by type with carbon green. (**C**) Interactions between the monophosphate of modeled 6F-MVAP and the surrounding amino acids in a superposition of the modeled *R. castenholzii* MPD, backbone atoms and carbon colored green, on the crystal structure of *S. epidermidis* MDD, backbone atoms and carbon colored gold. In *R. castenholzii* MPD, two divergent side chains, Arg83 and Lys161, putatively provide additional electrostatic interactions with the single phosphate group of MVAP (**3**). These amino acid side chains would clash with the second phosphate of MVAPP (**5**). These models suggest that the predicted active site topology of *R. castenholzii* MPD facilitates substrate recognition of MVAP (**3**) through complementary charged and polarized hydrogen bonds and excludes MVAPP (**5**) through steric incompatibility with its second phosphate.

## Discussion

The Chloroflexi are considered to be one of the oldest phyla of photosynthetic organisms (**Mulkidjanian et al., 2006**, **Hohmann-Marriott and Blankenship, 2011**). As photoheterotrophs, Chloroflexi use light for energy but cannot fix carbon dioxide as their primary source of carbon (**Bryant and Frigaard, 2006**). Of the six bacterial phyla reported to contain MVA pathway-bearing organisms, only the Chloroflexi contain IPK (**Lombard and Moreira, 2011**), but lack an obvious PMK. Although it appears from genome annotations that the Chloroflexi encode two incomplete branches of the essential MVA metabolic pathway, we demonstrate in one Chlorflexi bacterium that this MDD-like enzyme in fact acts catalytically as a bona fide MPD. MPD and IPK from this Chloroflexi bacterium, *R. castenholzii*, complete

**Table 4.** Active site phosphate-binding residues identified in MDDs across Archaea, Bacteria and Eukarya*

**MDD-like active site residues**

| *S. epidermidis* | Tyr18 | Lys21 | Ile27 | Ser139 | Ser141 | Ser192 |
|---|---|---|---|---|---|---|
| Bacteria | Tyr | Lys | X† | Ser | Ser | Ser |
| Sulfolobales | Tyr | Lys | Asn | Ser | Ser | Ser |
| Eukarya | Tyr | Lys | Ile/Asn | Ser | Ser | Ser |

**MPD-like active site residues**

| *R.castenholzii* | Tyr74 | Leu77 | Arg83 | Ala201 | Ser203 | Thr255 |
|---|---|---|---|---|---|---|
| Haloarchaea/Thermoplasmatales | Tyr /Phe | Met/Tyr | Arg | Ser | Ser | Polar‡ |
| Chloroflexi | Tyr | Leu | Arg/Thr | Ala | Ser | Thr |

*See **Table 4—source data 1**, **Table 4—source data 2**, **Table 4—source data 3**, and **Table 4—source data 4** for alignments.

†Not conserved among canonical bacterial MDDs.

‡Includes Glu, Asp, Asn, and Ser.

The following source data are available for table 4:

**Source data 1**. Amino acid sequence alignments of archaeal MDDs.

**Source data 2**. Amino acid sequence alignments of bacterial MDDs.

**Source data 3**. Amino acid sequence alignments of Chloroflexi MDDs.

**Source data 4**. Amino acid sequence alignments of eukaryotic MDDs.

the assembly of a hitherto unforeseen alternative MVA pathway that catalyzes sequential MVAP (**3**) decarboxylation and IP (**4**) phosphorylation resulting in the central hub metabolite of all organisms, the five-carbon isoprenoid building block IPP (**1**).

The consequences of the neofunctionalization of MDD to MPD are particularly noticeable in the substrate binding regions surrounding the diphosphate moiety of MVAPP (**5**) in canonical MDDs. The majority of the residues surrounding the diphosphate of MVAPP (**5**) in canonical MDDs are strictly conserved in structure- and sequence-based analyses (**Byres et al., 2007**). On the other hand, archaeal MDDs from Haloarchaea and Thermoplasmata contain MPD-like active site residues associated with MVAP (**3**) binding and turnover and their protein sequences cluster with the MPD from Chloroflexi (**Table 4**) (**Lombard and Moreira, 2011**). These MDDs belong to two archaeal classes that encode IPK but are missing PMK, suggesting that they also use this metabolic assembly of the alternative MVA pathway. The correlation between unique active site residues and alteration in substrate preference, MVAP (**3**) over MVAPP (**5**), among MPDs from Chloroflexi bacteria, reflects a lineage specific adaptation that allows for the utilization of the alternative MVA pathway; this adaptation may also apply to MPD-like MDDs of the archaeal phyla Haloarchaea and Thermoplasmata but awaits experimental verification currently in progress.

Despite key amino acid differences in the active site of the MDD-annotated *R. castenholzii* MPD, it nevertheless possesses high sequence similarity to canonical MDDs. Moreover, *R castenholzii* MPD and its closely related homologs annotated in several sequenced Chloroflexi species' genomes including, *Roseiflexus sp. RS-1*, *Herpetosiphon aurantiacus*, *Chloroflexus aggregans*, *Chloroflexus aurantiacus*, and *Chloroflexus sp. Y-400-fl* (http://www.genome.jp/kegg-bin/show_pathway?map00900), are all labeled as MDDs but share the alternative active site features of *R. castenholzii* MPD. Unexpectedly, this *R. castenholzii* gene, which is annotated as an 'MDD', catalyzes the phosphorylation-dependent decarboxylation of MVAP (**3**), thus serving as a long sought MPD requiring functional re-annotation in *R. castenholzii* and most likely the additional Chloroflexi species noted previously. It is possible that this enzyme has only recently emerged and undergone neofunctionaliation to acquire MPD activity from an MDD ancestor within the larger phylum of Chloroflexi heterotrophic photosynthetic bacteria. The emergence of this MPD activity may have also coincided with the loss of the *PMK* gene from

species within the narrower Chloroflexi class and/or acquisition of an *IPK* gene. Selection for such an alternative metabolic route to the core isoprenoid building block IPP may be due, in part, to a higher demand for metabolic flux through an alternative set of enzymes associated with the Chloroflexi MVA pathway. Indeed, the Chloroflexi class of bacteria appear to possess an expanded repertoire of downstream enzymes of isoprenoid metabolism suggesting a high demand for isoprenoid metabolites (http://www.genome.jp/kegg-bin/show_pathway?map00900).

While archaeal species contain an active IPK consistent with the presence of the alternative MVA pathway, most lack typical or atypical MDD genes; furthermore, a gene candidate capable of encoding an enzyme able to convert MVAP (**3**) to IP (**4**), the first postulated step of the alternative pathway, has not been identified. One candidate gene previously suggested to encode this decarboxylation activity is a gene encoding a dioxygenase-like protein that resides within the MVA operon in most Archaea (MJ0403 in *M. jannaschii*) (*Grochowski et al., 2006*). MJ0403 encodes a protein that is similar in sequence to subunit B of class III extradiol ring-cleavage dioxygenases and is also homologous to the human protein MEMO (mediator of erbB2-driven cell motility). Thus far, in vitro attempts to demonstrate decarboxylase activity for MJ0403 have failed to show any turnover of MVAP (**3**) to IP (**4**) even though the protein is quite easily produced heterologously.

While most archaea lack the MDD gene, certain species from the archaeal order Sulfolobales (including the archaeon *S. solfataricus*) encode a gene annotated as MDD. In our experiments, *S. solfataricus* PMK and MDD activities were detected in steady-state kinetic experiments and GC-MS assays, respectively (*Table 3*, *Figure 4*). A recent publication on the characterization of the classical MVA pathway enzymes from crude extracts of *S. solfataricus* supports the presence of an active classical MVA pathway in the order Sulfolobales, consistent with our in vitro results (*Nishimura et al., 2013*). While our experiments demonstrate that IPK is active in an in vitro assay, their experiments involving cell-free extracts of *S. solfataricus* indicate that IPK activity is undetectable. Nevertheless, these combined results apply to only a small set of Archaea, and the question remains as to how most archaeal species convert MVAP (**3**) to the essential metabolite, IPP (**1**).

Eukaryotic genomes examined to date appear to encode a classical MVA pathway with few exceptions (*Cassera et al., 2004*). We did however identify a spotty distribution of eukaryotes that contain IPK, and from those examined thus far, corresponding IPK activity that could be associated with an alternative route to IPP (**1**) through the MVA pathway. Since many eukaryotes encode putative enzymes of unknown function, they may also encode a cryptic MPD that would serve in an alternative biosynthetic route to IPP (**1**). All IPKs tested thus far were fully functional, demonstrating that true IPKs persist across most eukaryotic lineages, while some have been lost during rare evolutionary events, probably due to partial redundancy with the MVA pathway. The unusual phylogeny of IPK coupled with its membership in a family of kinases that phosphorylate such a broad range of substrates leave open the possibility that IPK may assume varied physiological roles, including phosphorylation of an IP-like substrate, IP recycling, and/or most notably, access to a pool of IPP through IP-IPP homeostatic control.

While the IPK gene is present within a spotty distribution of eukaryotes, the gene appears to be universally retained across the green plant lineage, which suggests that it plays a more universal role within the plant kingdom. Plants are unique in that they are currently known to encode two IPP-synthesizing pathways, the DXP pathway localized to the chloroplast and the MVA pathway localized to the peroxisome and cytoplasm (*Lange et al., 2000*, *Leivar et al., 2005*; *Nagegowda et al., 2005*; *Hsieh et al., 2008*). These pathways assume distinct metabolic roles within general isoprenoid biosynthesis (*Nes and Venkatramesh, 1999*; *Eisenreich et al., 2001*). It would not be surprising to identify yet another variation of the MVA pathway in the green plant lineage. In addition to IPP (**1**) recycling through IP (**4**), IPK would afford metabolic control of carbon availability from IP (**4**) to IPP (**1**). Noting that the diversity of primary and secondary isoprenoid products produced by plants often localize to subcellular compartments, organelles and specialized cell types (i.e. trichomes), an ubiquitous IPK in plants may regulate spatial and temporal control of isoprenoid diphosphate metabolism destined for a myriad of primary and specialized plant metabolites.

These results biochemically elucidate the terminal biosynthetic steps in an alternative or unconventional MVA pathway. The experiments described also definitively demonstrate metabolic routes to IPP (**1**) through the classical MVA pathway in previously overlooked organisms by including the kinetic characterization of IPKs, MDDs, MPDs, and PMKs taken from all three domains of life. Significantly, the studies presented provide the first experimental support for the existence of IPK catalytic activity in all three domains of life and a fully functional alternative MVA pathway in the photosynthetic heterotrophic bacterial class Chloroflexi.

# Materials and methods

## Cloning of IPK homologs

Archaeal IPK genes from *M. jannaschii*, *M. maripaludis* C5, and *S. solfataricus* P2, in addition to MDD and PMK genes from *S. solfataricus* P2, were cloned from genomic DNA from American Type Cell Cultures (ATCC, Manassas, VA) as previously described for *M. jannaschii* (**Dellas and Noel, 2010**) into a pET28a(+) vector containing a thrombin-cleavable N-terminal 8-His tag. IPK, MDD, and PMK genes from *A. thaliana*, *T. adhaerens*, *B. floridae*, and *R. castenholzii* were ordered as synthetic genes from Genscript (Piscataway, NJ, USA) and sub-cloned using Gateway technology from Invitrogen (San Diego, CA, USA) into pHIS9GW, an in-house pET28-based vector modified to contain a thrombin-cleavable 9-His tag.

## Protein expression and purification

All proteins were expressed according to a previously described procedure with several modifications (**Dellas and Noel, 2010**). Generally, each plasmid containing the gene of interest was transformed into BL21 (DE3) cells (Novagen®, Germany), grown at 37°C in 1 L cultures of TB media to an $OD_{600 nm}$ of 1.0, induced with 1 mM IPTG, and grown overnight at 20°C. All proteins were purified similarly and as previously described (**Dellas and Noel, 2010**); however, only the *M. jannaschii* protein was incubated at 80°C during purification. Additionally, the 9-His tag was removed from *R. castenholzii* MPD with thrombin for kinetic assays. MPD was then further purified by anion exchange on a Mono Q column (GE healthcare, Wauwatosa, WI) with a linear gradient of 0 M–1 M NaCl in 50 mM Tris-HCl, pH 8.0, over 40 column volumes and ultimately by size exclusion chromatography on Superdex 200 column (GE healthcare) developed in 50 mM Tris-HCl, pH 8.0, 0.5 M NaCl and 1 mM DTT.

## Steady-state kinetic analyses

Kinetic measurements were performed on IPKs from *M. maripaludis*, *S. solfataricus*, and *B. floridae* using a coupled pyruvate kinase–lactate dehydrogenase assay as previously described that employs IP concentrations ranging from 2 µM to 1 mM (**Dellas and Noel, 2010**). The substrate IP was purchased from Isoprenoids, LLC (>95% purity) (Tampe, FL). Steady-state kinetic curves were fitted using Prism (GraphPad Software Inc., San Diego, CA, USA) to compute $K_M$, $k_{cat}$, and where appropriate, $K_i$. Activity measurements were performed for *T. adhaerens* and *A. thaliana* using the coupled assay at four different IP concentrations (2 µM, 10 µM, 50 µM, and 100 µM) in triplicate.

Kinetic measurements were performed on MDDs from *B. floridae* and *A. thaliana* as discussed above with concentrations of (RS)-MVAPP (95% purity, Sigma, St. Louis, MO) ranging from 2 µM to 1 mM. Kinetic measurements were performed on MPD from *R. castenholzii* with concentrations of (RS)-MVAP (Sigma, 95% purity) ranging from 4 µM to 4 mM and PMKs from *B. floridae*, *A. thaliana*, and *S. solfataricus* with concentrations of (RS)-MVAP ranging from 5 µM to 2.5 mM.

## Fluorescence thermal shift assays

*R. castenholzii* MPD $T_m$s were calculated from pH 2.2 to 11.0 in 100 mM buffer (pH 2.2–3.8 citric acid; pH 4.0–4.8 sodium acetate; pH 5.0–5.8 sodium citrate; pH 6.0–6.8 sodium cacodylate; pH 7.0–7.8 sodium HEPES; pH 8.0–8.8 Tris-HCl; pH 9.0–11.0 CAPSO). Assays were carried out in white 96-well plates in a LightCycler 480 II (Roche Applied Science, Indianapolis, IN). Each well contained a 20 µl total volume, made up with 2 µl of 320 × SyproOrange Dye (Invitrogen, Carlsbad, CA) and 18 µl of MPD (2 µM) in 100 mM of the buffers listed above. The plate temperature was ramped from 20 to 99°C with 10 data points acquired per degree. SyproOrange dye (Invitrogen) was excited at 483 nm and fluorescence intensity (F) detected at 568 nm using the dynamic integration mode (max integration time, 999 ms). $T_m$s were obtained at temperatures (T) where the derivative of the thermograms (−dF/dT) was minimum.

## GC-MS assays

All GC-MS reconstitution assays were carried out in two steps. First, 1 µM of each of two enzymes (PMK and MDD for the classical MVA pathway or MDD and IPK for the alternative MVA pathway) was incubated for 30 min with 500 µM (RS)-MVAP, 4 mM ATP, and 10 mM $MgCl_2$ buffered with 50 mM Tris-HCl, pH 8.0. Second, 40 µl of this reaction was transferred to a glass vial containing 10 mM $MgCl_2$ buffered with 50 mM Tris-HCl, pH 8.0, containing at least a 150-fold excess of each of the following enzymes: *Escherichia coli* isopentenyl diphosphate isomerase (IPPI), *E. coli* farnesyl diphosphate

synthase (FPPS), and tobacco 5-*epi*-aristolochene synthase (TEAS). This reaction mixture was overlaid with ethyl acetate, incubated overnight, and vortexed to extract hydrocarbons from the aqueous layer the next day. Quantitative GC-MS analyses were performed as previously described (*Dellas and Noel, 2010*). All values were compared to a control reaction, where FPP was added in place of MVA pathway enzymes at appropriate concentrations to simulate complete turnover of MVAP.

## Bioinformatic analyses on IPK homologs

Public protein, cDNA, EST and genomic databases were searched for IPK homologs using individual IPK protein sequences, and profile Hidden Markov models built from several individual IPK clades. Genes were predicted from genomic sequences using Genewise (*Birney et al., 2004*) and TimeLogic GeneDetective (Active Motif Inc., Carlsbad, CA) programs, with manual editing. Protein sequences were aligned with Muscle (*Edgar, 2004*) and edited with ClustalX (*Larkin et al., 2007*) and JalView (*Waterhouse et al., 2009*). *Figure 2* was created using PhyML (*Guindon et al., 2005*) using the SPR model and rooted with fosfomycin kinase sequences. Manual editing was used to merge EST sequences and gene predictions, to correct frameshifts, and to fuse one gene split across two contigs. Discrepancies between individual ESTs were resolved to maximize sequence similarity to highly similar homologs.

IPK homologs from the Archaea domain were found in all but three of the 74 complete archaeal genomes found in the Integrated Microbial Genomes (IMG) database as of 8 Mar 2010 (*Markowitz et al., 2008*). The exceptions are *S. acidocaldarius* and *S. tokodaii*, and *Nanoarchaeum equitans*, a symbiont archaeon with a reduced genome.

Within the Bacteria domain, clear IPK homologs were only found in all five sequenced genomes of the class Chloroflexi, but not within other classes of the phylum Chloroflexi. Divergent homologs were found in *Streptomyces wedmorensis*, *Streptomyces fradiae* and one strain of *Pseudomonas syringae* (all probably fosfomycin kinases), and *Shewanella denitrificans*. The *P. syringae* gene is found only in a contig from strain PB-5123, and not several other sequenced strains. The sequence contains a frameshift within the ORF and lacks the H60 residue, both of which may be the result of sequencing errors.

In the Eukarya domain, searches for IPK homologs were made using the non-redundant amino acid (NRAA) Genbank database (*Benson et al., 2010*), the database of expressed sequence tags (dbEST) (*Boguski et al., 1993*), and a wide variety of genome databases, including those at Ensembl (www.ensembl.org) (*Birney et al., 2004*), Joint Genome Institute (JGI, genome.jgi-psf.org/), Baylor College of Medicine (www.hgsc.bcm.tmc.edu), Sanger Institute (www.genedb.org/) and the Broad Institute (www.broadinstitute.org). Searches were carried out with a series of IPK homologs (blastp against predicted peptides, tblastn against genome) using a hidden Markov model profile searched against the genome of interest using Gene Detective.

## MDD alignments

Alignments of MDDs were performed using a combined structure and sequence-based approach. Representatives from each of four groups (eukaryotes, archaea, bacteria, and Chloroflexi) were modeled using Swissmodel and superimposed to identify aligning active site residues (*Table 4*). Each of these four groups were then aligned with other sequences from the same group using the programs Muscle (*Edgar, 2004*) and Jalview (*Waterhouse et al., 2009*) to generate four separate alignments.

## Acknowledgements

Additional funding provided in part through the Arthur and Julie Woodrow Chair, a Joan Klein Jacobs and Irwin Mark Jacobs Senior Scientist Endowed Chair.

## Additional information

### Funding

| Funder | Grant reference number | Author |
|---|---|---|
| Howard Hughes Medical Institute | | Nikki Dellas, Suzanne T Thomas, Joseph P Noel |
| National Science Foundation | EEC0813570 | Nikki Dellas, Suzanne T Thomas, Joseph P Noel |

| Funder | Grant reference number | Author |
|---|---|---|
| National Science Foundation | MCB0645794 | Nikki Dellas, Joseph P Noel |
| National Institutes of Health | CA14195 | Gerard Manning |

The funders had no role in study design, data collection and interpretation, or the decision to submit the work for publication.

## Author contributions

ND, Conceived and completed the project including all experimental designs and implementations, deep phylogenetic analyses under the supervision of Gerard Manning, analyzed all data, drafted the complete article, and revised the article in response to reviewers' comments; STT, Carried out the measurements of the pH-dependent thermostability of MPD, carried out steady-state kinetic measurements on purified MPD with its initial affinity tag removed, and edited near final versions of the article; GM, Instructed Nikki Dellas on methods and interpretations of deep phylogenetic analyses described in the article, carried out computational methods described, provided an evolutionary context for the article, and together with Nikki Dellas drafted the initial article and edited the initially submitted article; JPN, Directed the project including experimental planning with Nikki Dellas, assisted in data analysis, interpreted data, drafted sections of the article together with Nikki Dellas and provided editing at all stages

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
