## [Decision Letter]

[Editors’ note: this article was originally rejected after discussions between the reviewers, but the authors were invited to resubmit after an appeal against the decision.]

Thank you for choosing to send your work entitled “Surprising functional divergence of the mevalonate pathway in all three domains of life” for consideration at *eLife*. Your article has been peer reviewed and we regret to inform you that your work will not be considered further for publication. Your submission has been evaluated by 3 reviewers, one of whom is a member of our Board of Reviewing Editors, and the decision has been discussed between the reviewers and with one of *eLife*'s Senior editors.

While all three reviewers found your work to be technically sound and the topic of the manuscript to be very interesting, they were split in their views on whether the manuscript reported discoveries that were of sufficient novelty or significance to merit publication in a general readership journal like *eLife*. After a group consultation among the reviewers, the emerging consensus opinion was that the manuscript was better suited for a specialty journal in the field of biochemistry or chemical biology. We hope the following specific comments will help you to revise the paper for submission to another journal.

*Reviewer #1*:

The manuscript by Dellas and colleagues describes the characterization of an alternative mevalonate pathway in Chloroflexi bacteria that inverts the canonical order of the terminal phosphorylation and decarboxylation steps in the conversion of MVAP to IPP. The authors determine that Chloroflexi possess an enzyme originally predicted based on sequence homology to be a mevalonate 5-diphosphate decarboxylase (MDD) that instead decarboxylates mevalonate 5-phosphate. This finding, combined with the existence of an isopentenyl phosphate kinase (IPK), establishes a complete mevalonate pathway in Chloroflexi. The authors proceed to make a convincing argument for discrete changes in the active site of the Chloroflexi decarboxylase that may account for the substrate switch from MVAPP to MVAP. The results reported in this manuscript certainly enrich our understanding of the phylogenetic diversity of mevalonate pathways, but this reviewer was not convinced that the findings are particularly surprising. The “enzyme repurposing“ discovered by the authors appears to reflect a rather simple tweak in substrate scope for the decarboxylase enzyme (i.e., MVAPP and MVAP only differ by a single phosphate group, so is it really unexpected that certain MDD-like enzymes might have evolved to switch substrate preference to MVAP over MVAPP?). Furthermore, as the authors note, there is already quite a bit of emerging literature on the general topic of diversification of mevalonate pathways, so this reviewer had difficulty judging the actual impact of the current manuscript. It is a solid and well-executed contribution, but appears to fall short of the high-impact study intended for a general readership journal like *eLife*.

*Reviewer #2*:

This manuscript uses a combination of bioinformatics and three-dimensional structures to discover a “missing“ enzyme in an alternate pathway to isopentenyl pyrophosphate (IPP) in the mevalonate pathway. The “usual“ pathway involves phosphorylation of mevalonate phosphate (MVAP) to the diphosphate (MVAPP) that then undergoes decarboxylation to IPP. However, in some organisms the kinase is “missing“; instead a kinase that phosphorylates isopentenyl phosphate (IP) could be identified. This suggested that these organisms would also have MVAP decarboxylase, producing IP that then is phosphorylated.

In this manuscript Noel and coworkers report that a protein that was predicted to be MVAPP decarboxylase is, in fact, MVAP decarboxylase, thereby providing the substrate for the previously chacterized IP kinase.

The work is carefully performed, with insights from three dimensional structures used to understand the structural basis for how/why the presumed MVAPP decarboxylase has an altered substrate specificity that allows the MVAP decarboxylase activity.

This manuscript can be accepted without change.

*Reviewer #3*:

The authors provide evidence for the identification of isopentenyl phosphate kinases (IPKs) in archaea, as well as a novel enzyme that catalyzes MVAP decarboxylation (MPD). These findings indicate the possibility of an exciting and novel alternative mevalonate-dependent biosynthetic route to isoprenoids, through MPD and IPK, rather than PMK and MDD. The authors used bioinformatic algorithms to identify IPK homologs (with a conserved active-site histidine), scattered throughout the domains of life (including animals). The authors purified 7 of these IPKS (one previously characterized homolog and 6 new) and confirmed their predicted enzymatic activity. The authors then enzymatically characterized PMKs and MDDs from organisms that had active IPKs. They found that one of the purified MDD homologues (from *R. castenholzii*) does not have MDD activity, but instead possesses an MPD activity in vitro. Single-pot multi-enzyme assays confirm the ability of these enzymes to synthesize isoprenoids. Not unexpectedly, the MDD-like RcMPD has amino acid changes and predicted structural changes from canonical MDDs that may explain a change in substrate specificity.

The major conclusions of this work are that functional IPKs are widespread and that MPDs exist and are utilized for IPP production-depend on in vitro studies with purified recombinant enzymes. The biological relevance of these findings would be greatly strengthened by additional experimental evidence that these enzymes function to produce isoprenoids within living cells (using, for example, cell feeding experiments such those described by Lange and Croteau 1999). In addition, kinetic characterization of the IPKs with additional substrates (such as mevalonate-5-phosphate, dimethylallyl monophosphate, and isopentenol) would increase confidence that isopentenyl monophosphate is indeed the preferred IPK substrate. The enzymatic characterization of RcMPD is key to the existence of the novel alternative pathway; however, the catalytic efficiency of this particular enzyme preparation was very low (2 logs less than the MDD enzymes in this study). Is there additional experimental evidence (for example, differential scanning fluorometry) that confirms that the purified enzyme is not denatured, resulting in loss of the canonical MDD activity? Perhaps MVAP is not actually the preferred natural substrate, given that the K_m_ is over 1 mM? Finally, confidence in the proposed enzymatic reactions would be increased by additional methods confirming the identity of the predicted products (for both IPKs and the RcMPD), other than linked enzymatic assays – the pyruvate kinase/LDH assays support ATP utilization and the FPS/TEAS assays support IPP production, but alternate products may be made. There are many analytical options (radiolabeled substrates/HPLC, LCMS, 31P NMR) that are typically used to confirm product identity.

[Editors' note: the following comments were sent to the authors upon evaluating the revised manuscript.]

Thank you for sending your work entitled “Unexpected Functional Divergence of the Mevalonate Pathway in All Three Domains of Life” for further consideration at *eLife*. Your article has been favorably evaluated by Deputy editor Detlef Weigel and 3 peer reviewers.

The Deputy editor and the other reviewers discussed their comments before we reached this decision, and the Deputy editor has assembled the following comments to help you prepare a revised submission.

Isoprenoids are essential metabolites for all living cells. Two main pathways are leading to isopentenyl diphosphate (IPP) and dimethylallyl diphosphate (DMAPP), the universal precursors of isoprenoids: the mevalonate (MVA) pathway in animals, fungi, Archaea, some eubacteria and in plant cytoplasm, and the methylerythritol phosphate pathway in most eubacteria, in the plant plastids as well as in some unicellular eukaryotes phylogenetically related to photosynthetic lineages. A variant of the MVA pathway has partially characterized in some Archaea with isopentenyl monophosphate (IP) as intermediates. This alternative MVA pathway requires an IP kinase (which has been characterized) and a putative MVA phosphate decarboxylase (which has not been characterized).

The manuscript submitted by Dellas, Manning, and Noel presents decisive results on the two above-mentioned enzymes. A fully alternative MVA pathway has been characterized for the first time in Choroflexi bacteria. The MVA phosphate decarboxylase, the missing link of this pathway, has been finally identified; its sequence has been compared to those of MVA diphosphate decarboxylase; and the biochemical activity has been demonstrated, indicating that the “putative MVA diphosphate decarboxylase” is finally a MVA monophosphate decarboxylase. The same data set is available for the IP kinase from Chloroflexi.

The most exciting contribution was the finding that a bacterial MVA diphosphate decarboxylase actually catalyzes the missing decarboxylation reaction proposed for the divergent MVA pathway, the decarboxylation of MVA phosphate. This is a very significant finding and is the strongest evidence to date for the existence of a mevalonate pathway with divergent last steps. The fact that this is the first report of such a pathway in bacteria is also very significant.

One wishes the authors could have provided in vivo proof of the operation of this pathway in bacteria or Archaea, but they do supply one of the next best things by demonstrating that they can combine the heterologously expressed mevalonate phosphate decarboxylase and isopentenyl monophosphate kinase to make the isopentenyl diphosphate (IPP) building blocks of terpenoid metabolism. Any additional data in this general direction that could be added in the revision would be most welcome.

---

## [Author Response]

[Editors’ note: the author responses to the first round of peer review follow.]

Reviewer #1:

*The manuscript by Dellas and colleagues describes the characterization of an alternative mevalonate pathway in Chloroflexi bacteria that inverts the canonical order of the terminal phosphorylation and decarboxylation steps in the conversion of MVAP to IPP. The authors determine that Chloroflexi possess an enzyme originally predicted based on sequence homology to be a mevalonate 5-diphosphate decarboxylase (MDD) that instead decarboxylates mevalonate 5-phosphate. This finding, combined with the existence of an isopentenyl phosphate kinase (IPK), establishes a complete mevalonate pathway in Chloroflexi. The authors proceed to make a convincing argument for discrete changes in the active site of the Chloroflexi decarboxylase that may account for the substrate switch from MVAPP to MVAP. The results reported in this manuscript certainly enrich our understanding of the phylogenetic diversity of mevalonate pathways, but this reviewer was not convinced that the findings are particularly surprising. The “enzyme repurposing“ discovered by the authors appears to reflect a rather simple tweak in substrate scope for the decarboxylase enzyme (i.e., MVAPP and MVAP only differ by a single phosphate group, so is it really unexpected that certain MDD-like enzymes might have evolved to switch substrate preference to MVAP over MVAPP?). Furthermore, as the authors note, there is already quite a bit of emerging literature on the general topic of diversification of mevalonate pathways, so this reviewer had difficulty judging the actual impact of the current manuscript. It is a solid and well-executed contribution, but appears to fall short of the high-impact study intended for a general readership journal like* eLife*.*

With all due respect to Reviewer 1, the repurposing of the enzyme in question, the decarboxylase of MVAPP, is totally unexpected and the simplicity of the explanation should not in any way suggest it is an obvious finding or only of incremental importance to isoprenoid metabolism. Clearly, the simplicity of the explanation strongly suggests that the decarboxylase in question, annotated as MVAPP decarboxylases (MDDs) in all available genome and transcriptome sequences requires both computational re-examination, and even more importantly, functional examination. Given the response of the audience at the recent international meeting on isoprenoid biology and metabolism where I (Noel) presented the results, many labs are already rethinking the currently accepted view of the mevalonate pathway, not only in Archaea where an alternative pathway was originally suggested to exist, but in all three domains of life. I would like to emphasize that the simplicity of this explanation of gene and enzyme neofunctionalization (or repurposing) is wholly unexpected and one of the major breakthroughs in the current submission.

Concerning the comment regarding the references we cite that review the emerging literature on the alternative mevalonate pathway, no studies to date have identified nor characterized the “missing” decarboxylase activity, enzyme or gene. I would like to again emphasize that the study described in the revised submission is the first discovery of this activity associated with an ubiquitous gene currently annotated as encoding conventional MDDs. Careful reading of these references in question demonstrate there is still much to learn.

Moreover, no one has ever suggested the presence of the IPK activity (the second major finding of the paper) in organisms outside of the Archaea (with the exception of the unfortunately erroneous paper mentioned below). We actually show convincingly in vitro that, in fact, this activity exists in all three domains of life, including throughout the green plant lineage and sporadically in metazoans including vertebrates up to the reptilians. Of course, there is much more to do particularly in vivo, and we have embarked on these experiments in plants and Chloroflexi. However, these in vivo studies are quite complex and will take time to develop the unconventional systems. We hope the reviewers and editors will agree that the in vivo work is well beyond the scope of the current manuscript. This manuscript, instead, describes for the first time the unexpected and critically important discovery of a major new metabolic route to isoprenoids in all organisms.

Finally, concerning the comment on the suitability of the paper for *eLife*, “It is a solid and well-executed contribution, but appears to fall short of the high-impact study intended for a general readership journal like *eLife*”, I again would respectfully disagree. Within the exploding field of metabolic research, the discovery of unexpected catalytic routes through essential metabolic pathways like the mevalonate pathway, that also hold incredible significance for disease (HMGR inhibition by statins), is of high-impact and general importance to many fields including metabolic engineering, synthetic biology, metabolic evolution and therapeutics.

Reviewer #2:

*This manuscript uses a combination of bioinformatics and three-dimensional structures to discover a “missing“ enzyme in an alternate pathway to isopentenyl pyrophosphate (IPP) in the mevalonate pathway. The “usual“ pathway involves phosphorylation of mevalonate phosphate (MVAP) to the diphosphate (MVAPP) that then undergoes decarboxylation to IPP. However, in some organisms the kinase is “missing“; instead a kinase that phosphorylates isopentenyl phosphate (IP) could be identified. This suggested that these organisms would also have MVAP decarboxylase, producing IP that then is phosphorylated*.

*In this manuscript Noel and coworkers report that a protein that was predicted to be MVAPP decarboxylase is, in fact, MVAP decarboxylase, thereby providing the substrate for the previously chacterized IP kinase*.

*The work is carefully performed, with insights from three dimensional structures used to understand the structural basis for how/why the presumed MVAPP decarboxylase has an altered substrate specificity that allows the MVAP decarboxylase activity*.

*This manuscript can be accepted without change*.

We respectfully concur.

Reviewer #3:

*The authors provide evidence for the identification of isopentenyl phosphate kinases (IPKs) in archaea, as well as a novel enzyme that catalyzes MVAP decarboxylation (MPD). These findings indicate the possibility of an exciting and novel alternative mevalonate-dependent biosynthetic route to isoprenoids, through MPD and IPK, rather than PMK and MDD. The authors used bioinformatic algorithms to identify IPK homologs (with a conserved active-site histidine), scattered throughout the domains of life (including animals). The authors purified 7 of these IPKS (one previously characterized homolog and 6 new) and confirmed their predicted enzymatic activity. The authors then enzymatically characterized PMKs and MDDs from organisms that had active IPKs. They found that one of the purified MDD homologues (from* R. castenholzii*) does not have MDD activity, but instead possesses an MPD activity* in vitro*. Single-pot multi-enzyme assays confirm the ability of these enzymes to synthesize isoprenoids. Not unexpectedly, the MDD-like RcMPD has amino acid changes and predicted structural changes from canonical MDDs that may explain a change in substrate specificity*.

*The major conclusions of this work are that functional IPKs are widespread and that MPDs exist and are utilized for IPP production-depend on* in vitro *studies with purified recombinant enzymes. The biological relevance of these findings would be greatly strengthened by additional experimental evidence that these enzymes function to produce isoprenoids within living cells (using, for example, cell feeding experiments such those described by Lange and Croteau 1999). In addition, kinetic characterization of the IPKs with additional substrates (such as mevalonate-5-phosphate, dimethylallyl monophosphate, and isopentenol) would increase confidence that isopentenyl monophosphate is indeed the preferred IPK substrate. The enzymatic characterization of RcMPD is key to the existence of the novel alternative pathway; however, the catalytic efficiency of this particular enzyme preparation was very low (2 logs less than the MDD enzymes in this study). Is there additional experimental evidence (for example, differential scanning fluorometry) that confirms that the purified enzyme is not denatured, resulting in loss of the canonical MDD activity? Perhaps MVAP is not actually the preferred natural substrate, given that the K*_*m*_
*is over 1 mM? Finally, confidence in the proposed enzymatic reactions would be increased by additional methods confirming the identity of the predicted products (for both IPKs and the RcMPD), other than linked enzymatic assays-the pyruvate kinase/LDH assays support ATP utilization and the FPS/TEAS assays support IPP production, but alternate products may be made. There are many analytical options (radiolabeled substrates/HPLC, LCMS, 31P NMR) that are typically used to confirm product identity*.

We are familiar with the paper in question. Unfortunately, Lange and Croteau had mis-identified the IPK activity. It turned out that Lange and Croteau had erroneously characterized the *ychB* enzyme from *E. coli*. This enzyme is one of the enzymes of the MEP (Rohmer) pathway of primary isoprenoid biosynthesis found in many, though not all bacteria, in *Plasmodium falciparum***,** and in plant chloroplasts. We previously worked on this enzyme structurally and functionally. In short, while the title suggests this 1999 paper is critical to the current study and evidence of an established alternative pathway over a decade ago, the *ychB* enzyme does not have a role in isopentenyl phosphate phosphorylation and therefore is not relevant to our study.

As mentioned in the response to Reviewer 1, we absolutely agree that more in vivo work must be carried out, but as argued above, this future work is well beyond the scope of the current findings as described in our current manuscript. The current study is rich with functional data from a phylogenetically diverse set of organisms. This will provide starting points for many labs working on phylogenetically diverse systems to probe the significance of these findings. In fact, we are in the midst of in vivo studies in Chloroflexi (establishing growth conditions and genetic manipulations) as well as in plants (*Arabidopsis thaliana* and *Nicotiana tabacum*). In the case of plants, some transgenics have been obtained and others are being established. Preliminary transcriptomic and metabolomic data in *Arabidopsis thaliana* show a strong correlation between IPK expression and the expression of other enzymes of downstream isoprenoid metabolism, i.e., plant steroids and sesquiterpenes as well as their small molecule products.

*In addition, kinetic characterization of the IPKs with additional substrates (such as mevalonate-5-phosphate, dimethylallyl monophosphate, and isopentenol) would increase confidence that isopentenyl monophosphate is indeed the preferred IPK substrate*.

The kinetic parameters for the IPK homologues characterized herein are in line with those observed for the archaeal IPKs that were originally characterized. While we agree with the reviewer (and have even suggested in the paper) that perhaps some of these enzymes (especially those in the Eukarya domain) may serve additional roles in vivo or are able to turn over a different substrate of natural or synthetic origin, we would argue that detailed use of a plethora of possible substrates in IPK assays is outside the scope of this paper given the sound kinetic data already obtained. As Poulter and our lab have shown, IPKs, when given alternative and mostly non-natural substrates, will phosphorylate them, but with kinetic constants not in line with in vivo flux through a bona fide metabolic pathway. Moreover, the kinetic constants obtained and presented in the current manuscript are quite reasonable and expected for highly specific IPKs and for a neofunctionalized MDD to MPD. In short, IPK enzymes from all three domains of life are able to carry out the IPK reaction with kinetic rates and K_M_s that are comparable to the enzyme that was originally characterized from Archaea.

*The enzymatic characterization of RcMPD is key to the existence of the novel alternative pathway; however, the catalytic efficiency of this particular enzyme preparation was very low (2 logs less than the MDD enzymes in this study). Is there additional experimental evidence (for example, differential scanning fluorometry) that confirms that the purified enzyme is not denatured, resulting in loss of the canonical MDD activity? Perhaps MVAP is not actually the preferred natural substrate, given that the K*_*m*_
*is over 1 mM*?

We acknowledge that the K_M_ of the enzyme is somewhat high; however, we would like to emphasize that the enzyme has virtually no measurable activity with MVAPP (its “natural” substrate based on the fact that it has high sequence homology to all other MVAPP decarboxylases) and a significant activity when assayed with MVAP. The in vivo work in progress will establish whether the K_M_ is “too” high or right in line with the steady-state concentrations of MVAP in Chloroflexi consistent with high flux through the alternative mevalonate pathway. It should be noted that smaller K_M_s often result in slower turnovers and that KMs for many enzymes of core metabolism are in the mM range often within 2-5-fold of the in vivo steady-state concentration of their substrates. Moreover, the existence of “metabolons” consisting of loosely associated complexes of consecutive enzymes in a metabolic pathway (such as in the purinosome) negate, to a large extent, the physical significance of steady-state KM measurements. Finally, careful examination of the Chloroflexi genomes all show the absence of the key kinase, phosphomevalonate kinase (PMK), which is essential for forming MVAPP. To date, we have not found IPK or other small molecule kinases from Chloroflexi that possess the PMK activity.

We have added a third paragraph in the Discussion to address these questions:

“Despite key amino acid changes in the active site of the annotated *Roseiflexus castenholzii* “MDD”, it nevertheless possesses high sequence similarity to canonical MDDs…”

*Finally, confidence in the proposed enzymatic reactions would be increased by additional methods confirming the identity of the predicted products (for both IPKs and the RcMPD), other than linked enzymatic assays-the pyruvate kinase/LDH assays support ATP utilization and the FPS/TEAS assays support IPP production, but alternate products may be made. There are many analytical options (radiolabeled substrates/HPLC, LCMS, 31P NMR) that are typically used to confirm product identity*.

We understand the reviewer’s concerns with regard to product identity; however, the assays are well established and can only operate with the substrates provided in vitro. Based on multi-enzyme GC-MS assays, we will only see a GC-detectable terpene product from the alternative pathway if *Rc*MPD is able to decarboxylate MVAP to IP, the latter of which is a substrate for IPK to make IPP. It should be noted that these are reconstituted in vitro reactions using highly purified components and not the result of using crude or even partially purified extracts possessing a mixture of other enzymes and metabolites that could lead to erroneous results. In short, if IP is not produced, IPK cannot produce IPP, and none of the downstream isoprenoids will be produced. It is also not surprising that *Rc*MPD performs a decarboxylation very similar to its closest homolog MDD, and the natural sequence variation in the *Rc*MPD active site are rationalized based on molecular mechanisms outlined in the text, listed in Table 3, and depicted in Figure 4. Finally, a very thorough functional study of all homologs from the entire Chloroflexi class that have genome sequences available is in progress, and product identity will be confirmed by a combination of recently obtained UPLC-MS and NMR instruments. These latter studies will provide a paper of record documenting a more detailed set of numbers for all the reactions. We feel that these studies are again beyond the scope of the current paper as this paper reports for the first time two major findings for the metabolism field: the presence of IPK-like activity across all three domains of life, and the first discovery and reconstitution of an alternative mevalonate pathway in vitro. Leaders in the field including Michel Rohmer (see above) have seen the results presented by Noel, and also have read the paper. Without bias or prejudice concerning our paper, all were shocked the original submission received a very lukewarm reception, particularly the suggestion that the combination of deep phylogenetic analyses coupled with quantitative in vitro kinetic analyses and reconstitution assays was not of high scientific impact.

[Editors’ note: the author responses to the re-review follow.]

*One wishes the authors could have provided* in vivo *proof of the operation of this pathway in bacteria or Archaea, but they do supply one of the next best things by demonstrating that they can combine the heterologously expressed mevalonate phosphate decarboxylase and isopentenyl monophosphate kinase to make the isopentenyl diphosphate (IPP) building blocks of terpenoid metabolism. Any additional data in this general direction that could be added in the revision would be most welcome*.

To specifically address the concerns of the reviewers, we have re-assayed the enzyme after thrombin cleavage of the 3 kDa affinity purification tag and found an order of magnitude improvement in the apparent K_M_, from 1.1 mM to 150 μM. In addition, Thermofluor stability assays demonstrate that recombinant MPD is extremely thermostable (expected as the Chloroflexi host lives optimally in hot springs at or slightly above the boiling point of water) with midpoint melting temperatures (Tm) above 90°C. Due to the low in vitro stability of components of the pyruvate kinases/lactate dehydrogenase assay and instrument limitations, *R. castenholzii* MPD could not be assayed at temperatures equivalent to the optimal growth conditions of the Chloroflexi host. We fully expect that the enzyme will perform more efficiently at temperatures approaching 100°C as observed for many enzymes from thermophilic organisms compared to their mesophilic counterparts. Unfortunately, the class of Chloroflexi in which new decarboxylase activity has been found are currently recalcitrant to genetic manipulation and difficult to grow in the laboratory without specialized fermentation systems. In the current revised manuscript, we now include the revised kinetic data along with a description of the assay for thermostability.

We also note that we have downplayed the second major discovery contained in the previous version of the manuscript, namely the clear biochemical identification of a fully functional and unanticipated IPK activity in all domains of life with a particular emphasis on the green plant lineage. We still include the biochemical results of in vitro IPK activity that is on par with previously measured activities in Archaea. However, we have significantly shortened and revised the current manuscript to focus attention on the major new quantitative discovery, the first characterization of a complete alternative MVA pathway through a newly discovered neofunctionalized mevalonate diphosphate decarboxylase to a mevalonate phosphate decarboxylase.

We are currently in the midst of studies of IPK's role in plant isoprenoid metabolism. Notably, using a series of knockouts and transient overexpression lines in the reference plants *Arabidopsis thaliana* and *Nicotiana tabacum*, we have accumulated preliminary evidence of a significant positive correlation between the expression of IPK activity in planta and 2-5-fold increases in plant sterol and sesquiterpene production. While these latter studies are still in progress and extend beyond the scope of the current manuscript, they lend further support for the unexpected discovery of alternative routes in plants for modulating economically important isoprenoid production through plant IPK activity.